# Structural and functional evidence of bacterial antiphage protection by Thoeris defense system via NAD$^+$ degradation

Donghyun Ka[1,4], Hyejin Oh[1,2,4], Eunyoung Park[1], Jeong-Han Kim[1,3] & Euiyoung Bae [1,3 ✉]

The intense arms race between bacteria and phages has led to the development of diverse antiphage defense systems in bacteria. Unlike well-known restriction-modification and CRISPR-Cas systems, recently discovered systems are poorly characterized. One such system is the Thoeris defense system, which consists of two genes, *thsA* and *thsB*. Here, we report structural and functional analyses of ThsA and ThsB. ThsA exhibits robust NAD$^+$ cleavage activity and a two-domain architecture containing sirtuin-like and SLOG-like domains. Mutation analysis suggests that NAD$^+$ cleavage is linked to the antiphage function of Thoeris. ThsB exhibits a structural resemblance to TIR domain proteins such as nucleotide hydrolases and Toll-like receptors, but no enzymatic activity is detected in our in vitro assays. These results further our understanding of the molecular mechanism underlying the Thoeris defense system, highlighting a unique strategy for bacterial antiphage resistance via NAD$^+$ degradation.

[1] Department of Agricultural Biotechnology, Seoul National University, Seoul 08826, Korea. [2] Department of Applied Biology and Chemistry, Seoul National University, Seoul 08826, Korea. [3] Research Institute of Agriculture and Life Sciences, Seoul National University, Seoul 08826, Korea. [4]These authors contributed equally: Donghyun Ka, Hyejin Oh. ✉email: bae@snu.ac.kr

Bacteriophages (phages) are viruses that infect bacteria[1]. They are the most abundant biological entities in the biosphere and coexist with their host[2]. The virus-rich environment and the constant exposure to viral infection led to the intense arms race between bacteria and phages[3,4], resulting in the development of diverse and sophisticated antiphage defense systems in bacteria[5]. Bacterial antiviral systems use various phage resistance mechanisms and function during distinct stages of the phage infection cycle[6,7]. Some of them have been thoroughly characterized for many years, including the restriction–modification and CRISPR-Cas systems[8,9]. Deciphering their detailed molecular mechanisms not only has contributed to our fundamental understanding of host–parasite interactions in microbiology but also has provided useful biotechnology tools such as restriction enzymes and Cas9 nuclease[10,11].

Systematic pangenomic analyses and subsequent experimental verification recently allowed the discovery of previously unknown bacterial defense systems[12–15]. Because antiphage defense genes tend to be clustered within specific loci termed "defense islands" in microbial genomes, candidate defense systems located near previously known defense genes were identified and functionally validated for their antiviral activities[12–15]. Among the newly discovered systems, the Thoeris defense system, named after a deity in Egyptian mythology, has been identified in more than 2000 microbial genomes with broad phylogenetic distribution[14]. It was detected in nine different taxonomic phyla, including a wide variety of bacteria and archaea[14].

The Thoeris system is composed of two genes, *thsA* and *thsB*. The first gene in the system, *thsA*, contains a domain (often annotated as sirtuin-like domain or macro domain) that binds to nicotinamide adenine dinucleotide (NAD) or its metabolites[16,17]. Sirtuins are a family of protein deacetylases whose activity is dependent on NAD hydrolysis[16]. They are widely distributed from bacteria to higher eukaryotes[18]. Its founding member, yeast Silent information regulator 2 (Sir2) is a histone deacetylase involved in a variety of cellular regulation[16]. CobB is a well-characterized bacterial sirtuin that regulates the function of acyl-CoA synthetase by lysine deacetlyation[19]. The macro domain is a widespread and conserved module of ~190 residues that can bind NAD metabolites or related molecules[17]. It is named after the C-terminal domain of macroH2A, a variant of histone H2A containing an extensive C-terminal nonhistone tail[20]. In several pathogenic bacteria, the connections between macro domains and pathogenesis have been suggested[21]. In some instances, *thsA* has an N-terminal transmembrane domain[14]. The second gene, *thsB*, contains a Toll/interleukin-1 receptor (TIR) motif and, in more than 50% of cases, is present in multiple, diverse copies around the single *thsA* gene[14]. Bacterial TIR-containing proteins have initially been hypothesized to function in pathogenesis but have also been implicated in other diverse mechanisms such as nucleic acid metabolism[22]. In several recent studies, the TIR domains of bacterial and eukaryotic proteins exhibited NAD$^+$ (the oxidized form of NAD) cleavage activities[23,24], suggesting that the antiphage function of the Thoeris system is related to NAD$^+$ binding and/or processing[14].

Two instances of the Thoeris system have been experimentally validated for their antiphage function[14]. One is from *Bacillus cereus* MSX-D12. The *thsA* gene and its downstream *thsB* gene conferred resistance against myophage infection when engineered into the model bacterium *Bacillus subtilis* BEST7003, which is devoid of an intrinsic Thoeris system[14]. *B. cereus* ThsA contains the sirtuin-like domain and lacks the N-terminal transmembrane domain and thus is predicted to be cytoplasmic[14]. Deletion experiment revealed that both ThsA and ThsB are required for the antiphage activity of the *B. cereus* Thoeris system[14]. The other validated Thoeris system is from *Bacillus amyloliquefaciens* Y2[14].

In this system, ThsA includes the macro domain with the additional N-terminal transmembrane domain and a shorter C-terminal region compared with the *B. cereus* homologue[14]. In the present study, we report structural and functional analyses of the ThsA and ThsB proteins from the *B. cereus* Thoeris defense system. Crystal structures of ThsA and ThsB were determined to resolutions of 2.5 and 1.8 Å, respectively. The enzymatic activities of the Thoeris proteins were also tested in the presence of NAD$^+$ and other nucleotides. We found that ThsA exhibited robust NAD$^+$ cleavage activity. Combined with previous findings, these results advance our understanding of the molecular mechanism underlying the Thoeris defense system and suggest that NAD$^+$ degradation is a previously unknown strategy for bacterial antiphage resistance.

## Results

**ThsA is an NAD$^+$-cleaving enzyme.** Since the Thoeris system was suggested to function via NAD$^+$ binding and hydrolysis[14], we performed analytical size-exclusion chromatography (SEC) to test the interaction between NAD$^+$ and purified recombinant *B. cereus* Thoeris proteins. We observed that the chromatographic peak of NAD$^+$ was divided into two parts in the presence of ThsA (Supplementary Fig. 1a), suggesting that ThsA processes NAD$^+$. We further assessed whether the Thoeris proteins can cleave NAD$^+$. NAD$^+$ was incubated with the *B. cereus* ThsA or ThsB, and the amount of remaining NAD$^+$ was measured at different time points using high-performance liquid chromatography (HPLC). In this NAD$^+$ cleavage assay, NAD$^+$ was rapidly degraded by ThsA, but not by ThsB (Fig. 1a), indicating that ThsA is an NAD$^+$-consuming enzyme. Using liquid chromatography–mass spectrometry (LC-MS), we identified nicotinamide (Nam) and adenosine-5′-diphosphoribose (ADPR) as products of NAD$^+$ degradation by ThsA (Fig. 1b and Supplementary Fig. 2), implying that ThsA cleaves NAD$^+$ into Nam and ADPR by hydrolyzing the Nam–ribosyl bond of NAD$^+$ (Fig. 1c). Notably, the NAD$^+$ cleavage activity was also observed in sirtuin proteins catalyzing NAD$^+$-dependent protein deacetylation, in which NAD$^+$ is used as a co-substrate[16]. The kinetic analysis revealed that the *B. cereus* ThsA exhibited a $K_m$ of 270 μM and a $k_{cat}$ of 33.9 min$^{-1}$ (Fig. 1d). The specificity constant ($k_{cat}/K_m$) of ThsA was calculated to be 2091 M$^{-1}$ s$^{-1}$. This falls within the range observed for NAD$^+$ cleavage by eukaryotic sirtuin proteins including the yeast Sir2 (3.5–8205 M$^{-1}$ s$^{-1}$)[25]. It is also noteworthy that the intracellular NAD$^+$ level of a bacterium *Escherichia coli* was previously estimated to be ~0.64 mM[26], which is within the same order of magnitude as the $K_m$ value of ThsA.

**ThsA contains sirtuin-like and Smf/DprA-LOG (SLOG)-like domains.** To further investigate the role of ThsA in Thoeris-mediated phage defense, we determined the crystal structure of *B. cereus* ThsA (Supplementary Table 1). The asymmetric unit contains four ThsA protomers (chains A–D), the structures of which are essentially identical. The root-mean-square deviation (RMSD) values of the Cα atomic positions between the protomers range from 0.5 to 0.8 Å. The protomers form two dimeric assemblies (AB and CD), which are very similar to each other (Supplementary Fig. 3a). The RMSD value of the Cα atoms between the two dimers is only 0.7 Å. The analysis using PISA[27] predicted tetrameric or octameric states formed with symmetry molecules ($A_2B_2$, $C_2D_2$, or $A_2B_2C_2D_2$) as probable quaternary structures of ThsA (Supplementary Fig. 3b). Consistent with this prediction, SEC with multi-angle light scattering (MALS) analysis confirmed that the tetrameric state of ThsA is predominant in a solution containing a trace amount of octamer (Supplementary Fig. 3c).

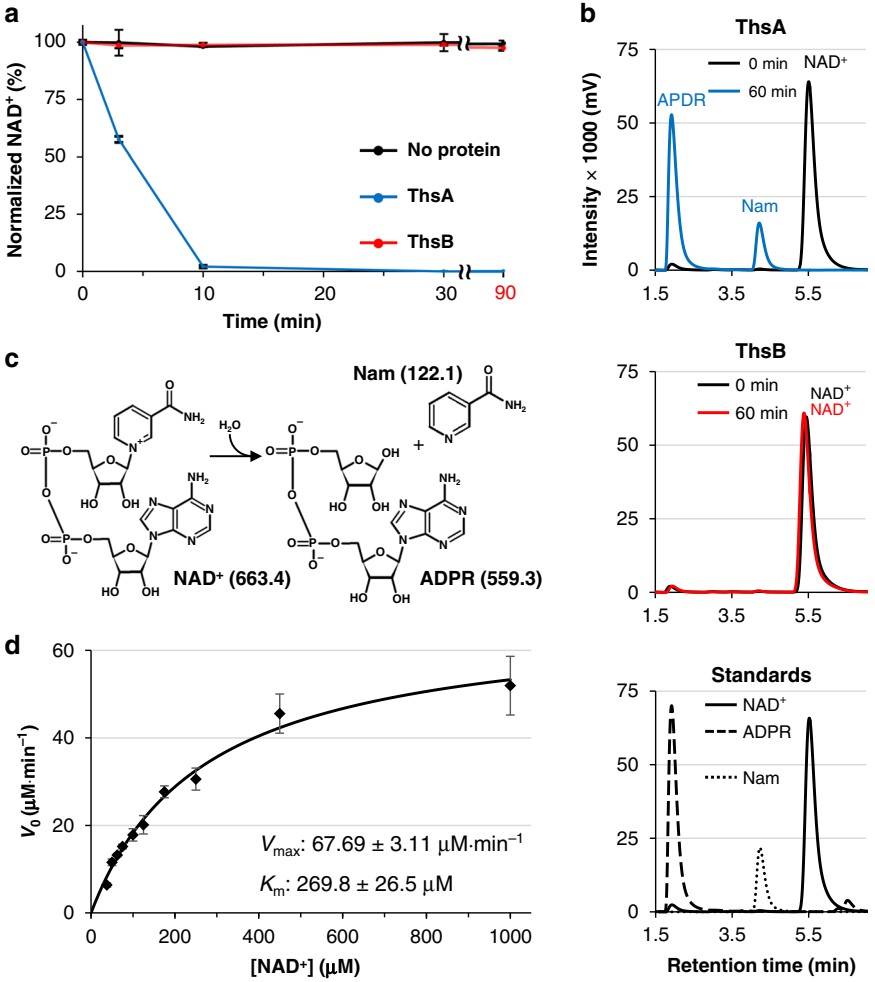

**Fig. 1 ThsA cleaves NAD$^+$ into Nam and ADPR. a** NAD$^+$ is degraded by ThsA, but not by ThsB. Data are presented as mean ± SEM for three independent experiments. **b** LC chromatograms of the NAD$^+$ cleavage products. MS analyses of each chromatographic peak are presented in Supplementary Fig. 2. **c** ThsA hydrolyzes the Nam–ribosyl bond of NAD$^+$. Molecular weights of NAD$^+$, Nam, and ADPR are indicated in parentheses. **d** Kinetic parameters of NAD$^+$ cleavage catalyzed by ThsA. Data are presented as mean ± SEM for four independent experiments. Source data are provided as Source Data file.

The protomer structure of ThsA revealed a two-domain architecture that consists of an N-terminal sirtuin-like domain (residues 1–283) and a C-terminal SLOG-like domain (residues 284–476) (Fig. 2a)[28]. The N-terminal domain contains the Rossmann-like fold, in which an extended seven-stranded parallel β-sheet (β3–β4–β2–β1–β5–β6–β7) is sandwiched between two helical layers (α1, α5, α6, and α11 on one side and α7–α10 on the other), comprising a three-layered globular structure (Fig. 2b). Three additional α-helices (α2–α4) form a small triangular structural module protruding from the Rossmann-like fold. This domain displays a structural similarity with sirtuin proteins, especially in the portion corresponding to the Rossmann-like fold (Fig. 2c). When the N-terminal domain of ThsA was structurally aligned with sirtuin proteins, the RMSD values ranged from 2.7 to 3.6 Å for ~170 Cα atoms (Supplementary Table 2). However, structural differences were also noted in the periphery of the Rossmann-like fold (Supplementary Fig. 4). The acetyl substrate-binding pocket of sirtuins is not available in the N-terminal domain of ThsA due to the blocking by its α7 helix (Supplementary Fig. 4), excluding the possibility that ThsA serves as an NAD$^+$-dependent deacetylase like sirtuins. ThsA also lacks the insertion element for zinc coordination present in sirtuins (Fig. 3a and Supplementary Fig. 4), the disruption of which results in loss of the NAD$^+$ hydrolysis function of sirtuins[16]. Thus, the N-terminal domain of ThsA does not

completely share the structural and functional characteristics of sirtuins.

The C-terminal domain of ThsA also possesses the Rossmann-like fold, which contains a five-stranded parallel β-sheet and three α-helices with a combination of βαβαβ (β8–α12–β9–α13–β10) and βαβ (β11–α16–β14) motifs (Fig. 2d). The residues connecting β11 and α16 form a β-hairpin structure. Additional α-helices (α14, α15, α17–α19) surround the edges of the β-sheet. A structural analysis indicated that this domain could be classified into the Sir2/TIR-associating SLOG (STALD) family within the SLOG superfamily[28]. Members of this protein family were predicted to function as sensors of nucleotides or related ligands, which are likely processed or modified by associating effectors[28]. The C-terminal domain of ThsA shares common features of the STALD family at the putative ligand-binding site, including secondary structural elements and conserved residues[28]. These residues include Ser288 and Phe357 in the loops following β8 and β10, respectively, and Arg371 and Glu403 of the helices (α15 and α16, respectively) in the ligand-binding pocket (Fig. 2e).

**NAD$^+$ cleavage by ThsA is linked to antiphage activity.** In a previous study, a point mutation (N112A) in ThsA resulted in complete loss of antiphage protection by the Thoeris system[14], indicating that Asn112 of ThsA plays a crucial role in the defense

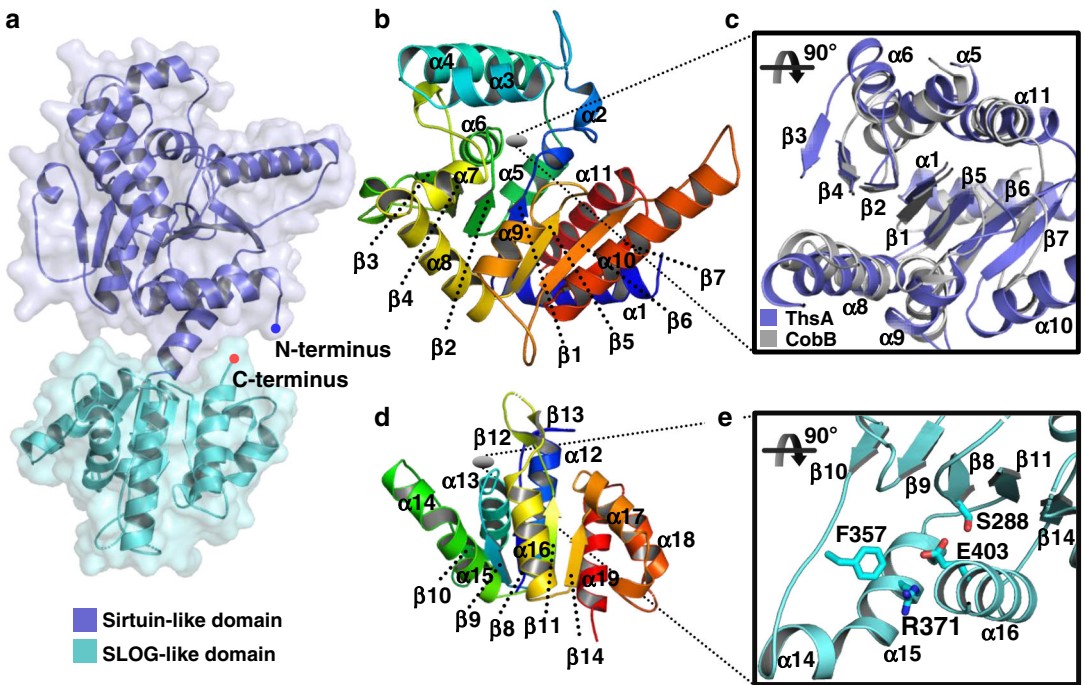

**Fig. 2 Crystal structure of ThsA. a** A two-domain architecture of ThsA consisting of an N-terminal sirtuin-like domain (blue) and a C-terminal SLOG-like domain (cyan). **b** Structure of the N-terminal sirtuin-like domain in ThsA. **c** Structural alignment of the Rossmann-like fold portion of the ThsA N-terminal domain (blue) with a sirtuin protein, *Thermotoga maritima* CobB (PDB ID: 2H2G; gray). **d** Structure of the C-terminal SLOG-like domain of ThsA. **e** Common features of the SLOG superfamily shared by the C-terminal domain of ThsA. ThsA residues conserved in the STALD family within the SLOG superfamily are shown in stick representation. Secondary structural elements of ThsA common to the SLOG superfamily are also indicated.

mechanism of Thoeris. In our crystal structure of ThsA, this residue is found within the putative $NAD^+$ binding site in its N-terminal domain (Fig. 3b and Supplementary Fig. 5). This strongly suggests a direct connection between the $NAD^+$ cleavage by ThsA and the protective function of the Thoeris system. To confirm this linkage, we generated ThsA mutants and tested their $NAD^+$ cleavage activities. Mutations were introduced at Asn112 and another residue in the putative $NAD^+$ binding pocket, His152 (Fig. 3b and Supplementary Fig. 5). These residues are also conserved in sirtuin proteins (Fig. 3c and Supplementary Fig. 6). In the $NAD^+$-bound sirtuin structure (Fig. 3b), the residues are located adjacent to the Nam–ribosyl portion of $NAD^+$, with the Asn and His side chains interacting with the Nam and ribose moieties, respectively. In sirtuins, the Asn residue is involved in substrate recognition[29,30], and the His side chain serves as a general base[31].

Both of the purified ThsA mutants, N112A and H152A, exhibited circular dichroism (CD) spectra similar to that of the wild type (Supplementary Fig. 7), which is characteristic of folded proteins. In analytical SEC, the chromatogram of the N112A mutant was essentially identical to that of the wild type control, but the H152A mutant was eluted significantly later (Supplementary Fig. 8a). These results suggest that the N112A mutant retains the overall folding of the wild type as well as its oligomeric state, but the H152A mutation disrupts the proper oligomerization of ThsA. In subsequent binding assays using analytical SEC, neither of the mutants bound $NAD^+$ (Supplementary Fig. 8b). In our enzyme assays, both N112A and H152A abolished the $NAD^+$ cleavage activity of ThsA (Fig. 3d).

Based on these observations, we concluded that the Asn112 residue is critical for $NAD^+$ hydrolysis of ThsA. For the H152A mutant, it cannot be ruled out that the inactivation is caused by the disruption of the functionally relevant oligomeric state. Thus, the same point mutation (N112A) that destroyed the

$NAD^+$-cleavage activity of ThsA resulted in complete inactivation of the Thoeris system[14]. Consequently, combined with the result from a previous study[14], our mutation analysis suggests that $NAD^+$ degradation by ThsA is linked to the antiphage function of the Thoeris defense system.

**ThsB exhibits structural similarity to TIR domain proteins.** The second gene in the Thoeris defense system, *thsB*, contains a TIR domain, and ThsB was previously implicated in $NAD^+$ hydrolysis[14]; several TIR domain proteins have been shown to exhibit $NAD^+$-cleavage activity[23,24]. However, in our in vitro assays, $NAD^+$ was cleaved by ThsA, but not by ThsB (Fig. 1a). In the TIR $NAD^+$ hydrolases, $NAD^+$-cleavage activity was found to be dependent on self-association[32,33]; mutations disrupting functionally relevant oligomeric states abolished the activity[32,33] and addition of macromolecular crowding agents simulating the intracellular environment stimulated $NAD^+$ degradation[32]. By contrast, our SEC-MALS analysis indicated that ThsB exists as a monomer even at high concentrations (~1 mM; Supplementary Fig. 9). No stimulating effect by a macromolecular crowding agent (PEG 400) was observed for ThsB (Supplementary Fig. 10). It is also unlikely that ThsB alters the function of ThsA or catalyzes the potential pathway(s) downstream of $NAD^+$ cleavage because ThsA was still active in the presence of ThsB, which did not further process the cleavage products Nam and ADPR (Supplementary Fig. 11). We also suspected the possibility that ThsB may function as a component of a larger multi-protein complex formed with ThsA. However, in our analytical SEC experiment, ThsB was eluted separately from ThsA with and without $NAD^+$ (Supplementary Fig. 1b, c). When expressing $(His)_6$-maltose binding protein (MBP)-tagged ThsA together with untagged ThsB in *E. coli* cells, ThsB did not co-purify with ThsA in nickel-affinity chromatography (Supplementary Fig. 12). These

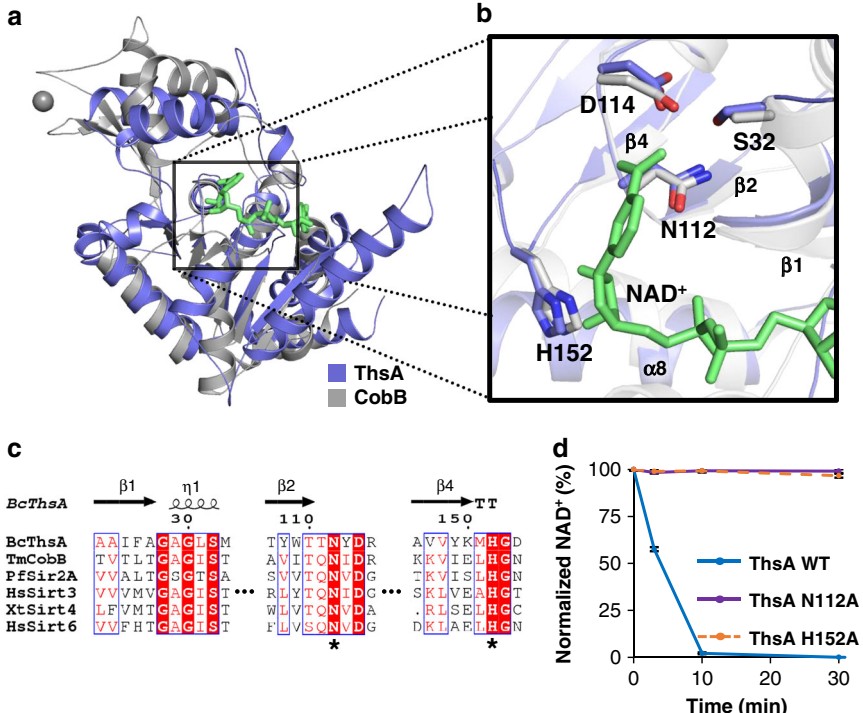

**Fig. 3 NAD$^+$ cleavage activity is intrinsic to the N-terminal sirtuin-like domain of ThsA. a** Structural alignment of the ThsA N-terminal domain (blue) with a sirtuin deacetylase, *T. maritima* CobB (PDB ID: 2H2F; gray) bound to a co-substrate, NAD$^+$ (green). The essential zinc ion in the CobB structure is represented as a gray sphere. **b** Close-up view of the NAD$^+$ binding site. Asn112 and His152 of ThsA align with sirtuin residues interacting with NAD$^+$. **c** Sequence alignment of the N-terminal domain of ThsA with sirtuin proteins. Asn112 and His152 of ThsA are marked with asterisks. White character in red box indicates strict identity. Red character and blue frame represent similarity in a group and across groups, respectively. Alignment of the full-length amino acid sequences is presented in Supplementary Fig. 6. **d** Asn112 and His152 of the N-terminal sirtuin-like domain are critical for NAD$^+$ hydrolysis of ThsA. N112A and H152A mutations of ThsA abolished NAD$^+$ cleavage activity of ThsA. The graph for the wild-type ThsA (Fig. 1a) is shown again as a control for comparison. Data are normalized to the values measured at 0 min for each protein and presented as mean ± SEM for three independent experiments. Source data are provided as Source Data file.

observations suggest that ThsB does not form a stable complex with ThsA.

To gain structural insight into the role of ThsB in the Thoeris defense system, we determined the crystal structure of *B. cereus* ThsB (Supplementary Table 1). Two ThsB molecules were found in the asymmetric unit, and the RMSD value of the Cα atoms was only 0.9 Å, indicating high similarity. ThsB exhibited a four-layered structure in which α-helices and β-sheets were alternately stacked (Fig. 4a). A five-stranded parallel β-sheet (β2–β1–β3–β4–β10) and five α-helices comprise the Rossmann-like fold, with α1, α2, and α5 on the concave side of the β-sheet and α3 and α4 on the convex side (Fig. 4b). An additional antiparallel β-sheet (β5–β9) is located adjacent to the α4 helix in an approximately parallel orientation (Fig. 4c).

A search for structural neighbors using the Dali server[34] did not identify a significant match to the entire ThsB structure. However, excluding the additional antiparallel β-sheet, the Rossmann-like fold portion of ThsB revealed a structural resemblance to TIR domain proteins such as human sterile alpha and TIR motif containing 1 (SARM1), nucleoside monophosphate hydrolases, and toll-like receptors (Supplementary Table 3 and Fig. 4d). SARM1 is an NAD$^+$-cleaving TIR family protein[32]. Despite the high structural similarity to SARM1, ThsB did not cleave NAD$^+$ in our in vitro assay (Fig. 1a). If ThsB is a nucleoside monophosphate hydrolase, the conservation of Phe6 in the potential active site strongly suggests that ThsB would prefer substrates containing 2′-ribosyl groups to those containing 2′-deoxyribosyl moiety (Supplementary Fig. 13a, b)[35]. On this basis, we tested the hydrolase activity of ThsB in the presence of

various ribonucleotides such as AMP, GMP, UMP, and CMP, but no hydrolysis was detected (Supplementary Fig. 13c). This raises other possibilities; the enzymatic activity of ThsB may be triggered by an unknown in vivo mechanism, or ThsB may have nonenzymatic function(s) as a toll-like receptor.

## Discussion

In this study, we demonstrated experimentally that the ThsA protein in the Thoeris defense system is an NAD$^+$-cleaving enzyme. This is consistent with the previous proposition that NAD$^+$ binding and hydrolysis are essential for Thoeris-mediated antiphage resistance[14]. Our mutation analysis also revealed a direct connection between NAD$^+$ cleavage by ThsA and the protective activity of the Thoeris system. The N112A point mutation of ThsA, which was shown to neutralize the Thoeris defense system in a previous plaque assay[14], abolished the NAD$^+$ hydrolase activity of ThsA in our experiment (Fig. 3d). These results suggest that the Thoeris system uses NAD$^+$ degradation as an antiphage strategy, indicating a previously unknown bacterial defense mechanism against phage infection. It cannot be completely ruled out that the binding to NAD$^+$, not its cleavage, is important for the antiphage function since the N112A mutant did not bind NAD$^+$. To further study structural aspect of NAD$^+$ cleavage by ThsA, we tried to crystallize NAD$^+$-bound complexes of both wild-type and mutant ThsA proteins, but our attempts were not successful.

In addition to the sirtuin-like N-terminal domain, ThsA contains a C-terminal SLOG-like domain, the function of which is less well understood. SLOG superfamily proteins are involved in

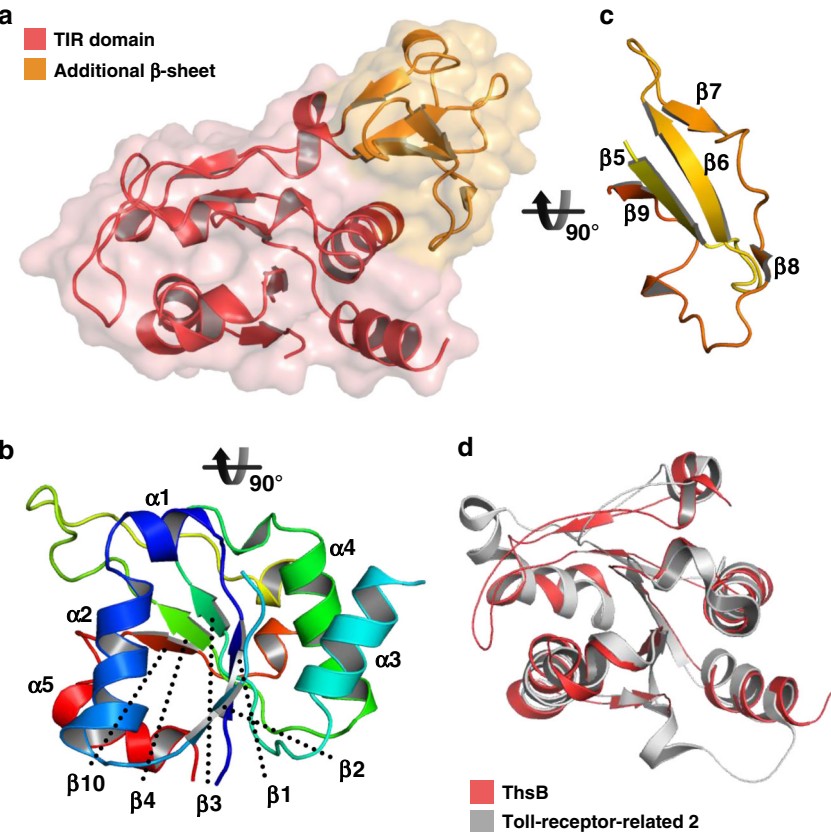

**Fig. 4 Crystal structure of ThsB. a** Overall structure of ThsB containing a TIR domain (red) and an additional β-sheet (orange). Structures of the TIR domain exhibiting the Rossmann-like fold (**b**) and the additional antiparallel β-sheet (**c**) of ThsB. **d** Structural alignment of the TIR domain of ThsB (red) with *Hydra vulgaris* toll-receptor-related 2 (PDB ID: 4W8G; gray).

various nucleotide-related processes[28], including the modification of nucleotides for cytokinin hormone production in plants[36,37], interaction with single-stranded DNA during bacterial transformation[38], and binding to molybdenum cofactors, which often share structural similarities to nucleotides[39]. *Streptococcus pneumonia* DprA and its *B. subtilis* ortholog (Smf) belong to the SLOG superfamily and have been previously shown to bind single-stranded DNA of random sequence[38]. However, the DNA binding affinity was not observed for ThsA in our electrophoretic mobility shift assay (EMSA; Supplementary Fig. 14). In a previous bioinformatics study, certain versions of SLOG-like domains were identified as components of nucleotide-centric systems involved in biological conflicts and were predicted to serve as sensors recognizing nucleotides or the enzymes that process them[28]. Recently, *B. subtilis* YpsA, a bacterial member of the SLOG superfamily, was implicated in the response to oxidative stress and regulation of cell division[40]. It is also noteworthy that ~30% of ThsA homologues do not contain the SLOG-like domain but rather an additional N-terminal multi-transmembrane domain[14]. These findings suggest that the SLOG-like domain of ThsA is involved in the recognition and/or transmission of nucleotide-related or other molecular signals. Thus, the C-terminal domain of ThsA may play a role in the regulation or signaling associated with the NAD$^+$ cleavage activity of the N-terminal domain. Nevertheless, the exact function of the SLOG-like domain in the Thoeris defense system awaits further investigation.

In the structural and functional analyses, we could not establish a role for ThsB in the Thoeris defense system. ThsB possesses a TIR domain and was originally predicted to be an NAD$^+$-cleaving enzyme[14], since several TIR domain-containing proteins show NAD$^+$ cleavage activity[23,24]. However, ThsB did not exhibit

NAD$^+$ cleavage activity in our experiment (Fig. 1a and Supplementary Fig. 10). It is possible that the enzymatic activity of ThsB is regulated by an unidentified in vivo mechanism, or we failed to recreate functionally relevant reaction condition(s) for ThsB in our in vitro assays. Alternatively, ThsB may have a function different from NAD$^+$ cleavage since ThsA is responsible for cleaving NAD$^+$ in the Thoeris system. Based on the presence of *thsB* in multiple, diverse copies in more than 50% of Thoeris systems, it was previously proposed that ThsB participates in recognition of phage infection with various ThsB proteins sensing different phage components[14]. In *B. cereus* MSX-D12, the second *thsB* gene is found upstream of the pair of *thsA* and *thsB* genes characterized in this study. The amino acid sequence identity is only 18% between the two ThsB homologues, indicating the diversity of multiple ThsB copies in Thoeris defense system. Notably, the single *B. cereus thsA* gene and its downstream *thsB* gene were sufficient to confer antiphage resistance[14], suggesting the dispensability of the upstream *thsB* gene.

NAD$^+$ is an important metabolic component in all forms of life, and its cleavage has previously been implicated in various biological conflicts. *Staphylococcus aureus* TirS virulence factor induces NAD$^+$ loss in mammalian cells[23]. *Mycobacterium tuberculosis* secretes NAD$^+$-hydrolyzing toxin to trigger macrophage necroptosis[41]. The plant commensal bacterium *Pseudomonas protegens* employs NAD$^+$-degrading effectors that exhibit interbacterial antagonism[42]. In plants, NAD$^+$ cleavage, triggered by the recognition of pathogen effectors, induces localized cell death through a yet unknown downstream mechanism to restrict pathogen infection[32,33]. NAD$^+$ depletion also mediates local axonal degeneration after injury as an intrinsic self-destruction program[24,32,43]. Based on these observations, we carefully raise

the possibility that the Thoeris system is an "altruistic suicide" defense system that kills phage-infected cells via NAD$^+$ degradation to prevent parasite transmission to neighboring bacteria. It is unclear whether the NAD$^+$ cleavage by the Thoeris system is sufficient to cause cell death by NAD$^+$ depletion. During the expression of recombinant ThsA and/or ThsB in *E. coli*, we did not observe growth arrest or toxicity. This suggests that the Thoeris system does not deplete the intracellular NAD$^+$ to drive direct cell death and is a part of a more elaborate defense mechanism requiring downstream signaling after NAD$^+$ cleavage as in the case of the plant TIR NAD$^+$ hydrolases[32,33]. The more precise molecular mechanism of bacterial antiphage protection involving the Thoeris defense system remains to be determined, including the regulation of ThsA and the function of ThsB.

## Methods

**Cloning, expression, and purification.** Synthetic *thsA* and *thsB* genes of *B. cereus* MSX-D12 were cloned into pET28a vectors with an N-terminal (His)$_6$-MBP tag and a tobacco etch virus protease cleavage site. Mutant genes were generated by polymerase chain reaction using mismatched primers (Supplementary Table 4). *E. coli* BL21(DE3) cells (Enzynomics) transformed with these constructs were cultured in LB medium at 37 °C until the optical density at 600 nm reached 0.7. Protein expression was induced by the addition of 0.5 mM isopropyl-β-D-thioga-lactopyranoside (IPTG), followed by incubation at 17 °C for 16 h. The cells were harvested by centrifugation and resuspended in lysis buffer (300 mM NaCl, 10% (w/v) glycerol, 5 mM β-mercaptoethanol, 0.3 mM phenylmethylsulfonyl fluoride, 0.3 mM Triton X-100, 20 mM Tris-HCl pH 7.5).

After sonication and centrifugation, the supernatant was loaded onto a 5-mL HisTrap HP column (GE Healthcare) pre-equilibrated with purification buffer (300 mM NaCl, 10% (w/v) glycerol, 5 mM β-mercaptoethanol, 30 mM imidazole, 20 mM Tris-HCl pH 7.5). After washing the column with purification buffer, the bound proteins were eluted by applying a linear gradient of imidazole (up to 450 mM). The (His)$_6$-MBP tag was cleaved via the tobacco etch virus protease and separated using a 5-mL HisTrap HP column (GE Healthcare). Proteins were further purified using the HiLoad 16/60 Superdex 200 column (GE Healthcare) equilibrated with buffer (200 mM NaCl, 2 mM dithiothreitol (DTT), 20 mM HEPES pH 7.5).

**Enzyme activity assay.** Proteins (2.5 μM) were incubated with 250 μM substrate (NAD$^+$, AMP, GMP, UMP, and CMP) in 100 μL buffer (63 mM NaCl, 14 mM Tris-HCl pH 7.5) at 25 °C. Reactions were stopped by the addition of 100 μL of ice-chilled 1 M HClO$_4$, followed by neutralization with 33.4 μL of 3 M K$_2$CO$_3$ on ice. After centrifugation of the sample, 99 μL of the supernatant was mixed with 11 μL of 0.5 M potassium phosphate buffer (pH 7.0) for further analysis.

**HPLC.** Reaction products were analyzed using the 1100 HPLC system (Agilent) with a diode array detector at 260 nm. Sample (5 μL) was loaded onto a YMC-Triart C18 column (100 × 2.0 mm, S-3 μm, 12 nm; YMC) equilibrated with mobile phase A (0.1% (v/v) formic acid and 5 mM ammonium formate in water). Separation was performed at a flow rate of 0.2 mL min$^{-1}$ using the gradient program for mobile phase B (0.1% (v/v) formic acid and 5 mM ammonium formate in methanol): 0–10 min, 0%; 13 min, 50%; 16–20 min, 95%; 25–40 min, 0%.

**LC-MS.** LC-MS was performed using the Nexera X2 UHPLC system coupled to the LCMS-8040 triple quadrupole mass spectrometer (Shimadzu). LC was conducted using the same column and the conditions described for HPLC with the exception of the gradient program for mobile phase B, which was as follows: 0–3 min, 0%; 10 min, 50%; 13–15 min, 95%; 18–23 min, 0%. In the MS system, ionization of target analytes was performed in the electrospray ionization positive mode. Mass spectra of the samples were obtained by scanning between *m/z* 100 and 800. The desolvation line and heat block temperature were 250 and 400 °C, respectively. The nebulizing (nitrogen) and drying gas (nitrogen) flow were 3 and 15 L min$^{-1}$, respectively. As LC-MS software, LabSolutions (ver. 5.60; Shimadzu) was used for data processing.

**Determination of enzyme kinetic parameters.** Kinetic parameters ($K_m$ and $V_{max}$) were determined using 2.0 μM ThsA based on the initial velocity measurement of NAD$^+$ consumption during the first 60 s of the reaction. The remaining NAD$^+$ amount was measured using HPLC as described above. Data were fitted to the Michaelis–Menten equation using SigmaPlot (Systat Software).

**Crystallization and structure determination.** The selenomethionyl proteins were expressed in *E. coli* BL21 (DE3) cells grown in minimal medium supplemented with selenomethionine (SeMet)[44]. The cells transformed with the pET28a vectors containing the *thsA* and *thsB* genes were cultured in M9 medium at 37 °C until the optical density at 600 nm reached 0.6. To inhibit Met biosynthesis, the culture was

supplemented with other amino acids (0.5 mM Lys, 0.8 mM Val, 0.8 mM Thr, 0.6 mM Phe, 0.8 mM Leu, 0.8 mM Ile). After incubation at 17 °C for 25 min, the cells were further supplemented with 0.25 mM SeMet. Protein expression was induced by the addition of 0.5 mM IPTG, followed by incubation at 17 °C for 20 h. The sele-nomethionyl proteins were purified as the native proteins described above. The incorporation of SeMet was monitored by mass spectrometry. Seven and two Met residues out of eight and two possible sites were substituted with SeMet for ThsA and ThsB, respectively. ThsA crystals were obtained at 20 °C by the sitting-drop vapor diffusion method from 43 mg mL$^{-1}$ protein solution in buffer (200 mM NaCl, 7 mM DTT, 20 mM HEPES pH 7.5) mixed with an equal amount of reservoir solution (22% (v/v) polypropylene glycol 400, 0.2 M MgCl$_2$, 0.1 M MES pH 6.5). ThsB crystals were obtained at 20 °C by the sitting-drop method from 23 mg mL$^{-1}$ protein solution in buffer (200 mM NaCl, 2 mM DTT, 20 mM HEPES pH 7.5) mixed with an equal amount of reservoir solution (3 M NaCl, 0.1 M Tris pH 8.5). The crystals were flash-frozen in liquid nitrogen without additional cryoprotecting reagents. Diffraction data were collected at the beamline 7A of the Pohang Accelerator Laboratory at 100 K. Diffraction images were processed using HKL2000[45]. The determinations of selenium positions, density modification, and initial model building were conducted using PHENIX[46]. The structures were completed using alternate cycles of manual fitting in COOT[47] and refinement in PHENIX and REFMAC5[48]. The stereochemical quality of the final models was assessed using MolProbity[49].

**Analytical SEC.** Analytical SEC was performed using the Superdex 200 10/300 GL column (GE Healthcare). The column was equilibrated with buffer (150 mM NaCl, 1 mM DTT, 20 mM HEPES pH 7.5). Proteins (20 μM) and NAD$^+$ (40 μM) were incubated at 4 °C for 1 h and loaded onto the column at a flow rate of 0.5 mL min$^{-1}$. Elution fractions were analyzed by sodium dodecyl sulfate–polyacrylamide gel electrophoresis and visualized by Coomassie staining.

**SEC-MALS.** SEC-MALS analysis was performed on the Superdex 200 Increase 10/300 GL column (GE Healthcare) coupled to the DAWN HELEOS II (18-angle) and Optilab T-rEX instruments (Wyatt Technology). The column was equilibrated with buffer (200 mM NaCl, 2 mM DTT, 20 mM HEPES pH 7.5), after which ThsA (11 mg mL$^{-1}$) and ThsB (24 mg mL$^{-1}$) were loaded onto the column at a flow rate of 0.5 mL min$^{-1}$ at 25 °C. Data were analyzed using ASTRA 6 software (Wyatt Technology).

**CD spectroscopy.** CD spectra were measured with protein samples (0.8 μM) in 500 μL buffer (10 mM potassium phosphate pH 7.5) at 25 °C using a J-815 CD spectropolarimeter (Jasco).

**EMSA.** DNAs (0.5 μM) were incubated with proteins (2.5 μM) and NAD$^+$ (1 mM) in reaction buffer (100 mM NaCl, 20 mM HEPES pH 7.5) at 20 °C for 30 min. The samples were analyzed on 9% acrylamide gels and visualized by ethidium bromide staining.

**Reporting summary.** Further information on research design is available in the Nature Research Reporting Summary linked to this article.

## Data availability

The atomic coordinates and structure factors for ThsA and ThsB were deposited in the Protein Data Bank[50] with the accession codes 6LHX [https://doi.org/10.2210/pdb6LHX/pdb] and 6LHY [https://doi.org/10.2210/pdb6LHY/pdb], respectively. The source data underlying Figs. 1a, b, d and 3d and Supplementary Figs. 1a–c, 3c, 7–11a, 12a, b, 13c and 14a are provided as Source Data file. Other data are available from the corresponding author upon reasonable request. Source Data are provided with this paper.

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

## Acknowledgements

We thank the staff of the beamline 7A of the Pohang Accelerator Laboratory for their support with the data collection, Gyujin Lee for assistance in activity assay, and Jasung Koo for help in CD spectroscopy. This work was supported by the National Research Foundation of Korea (NRF) grant funded by the Korea government (MSIT) (No. 2019R1A2C1086298) and the BK21 Plus Program of the Department of Agricultural Biotechnology, Seoul National University.

## Author contributions

D.K. and E.B. conceived the study. D.K. and H.O. performed the protein purification, crystallization, enzymatic assays, SEC, and EMSA. D.K. determined the crystal structures. D.K., H.O., E.P., and J.-H.K. conducted the HPLC and LC-MS analyses. H.O. performed CD spectroscopy. D.K., H.O., and E.B. analyzed the data, and wrote the paper with input from all authors.

## Competing interests

The authors declare no competing interests.
