## [Peer Review File · Nature Communications]

Reviewers' comments:

Reviewer #1 (Remarks to the Author):

In this work Ka et al. determine the crystal structures of the ThsA and ThsB proteins from the Thoeris anti-phage defense system of *Bacillus cereus*. They confirm that the ThsA protein has NAD⁺ cleavage activity in vitro, as was previously postulated by the Sorek group. They were unable to detect in vitro enzymatic activity for ThsB. Overall this study provides molecular detail of the proteins that comprise the Thoeris system in *B. cereus*. It is increasingly apparent that anti-phage defense systems are playing a very important role in the evolutionary battles between bacteria and their viruses. Further, evidence of the parallels and connections between eukaryotic and bacterial innate immune systems is mounting. Thus, this work is timely and likely to be of interest to a wide audience.

The manuscript is clearly written and easy to follow. However I found it too focused on only the structure and lacking in the discussion of the implications for the biology of the system. I also found discussion of the conserved protein domains too focused on examples from eukaryotes. The examples used should be bacterial whenever possible as the target audience is phage and bacterial experts.

Thoeris system is found in >2000 microbial genomes –how diverse are the bacteria it's found in? Is it only strains of *Bacillus*, or is it widely distributed across different bacterial genus?

In some cases ThsA has a transmembrane domain – how would this play into its activity? Where else are sirtuin (or Macro) domains found? In bacteria?

The discussion around the ThsB protein should be fleshed out. In some cases, there are multiple ThsB proteins found around a single ThsA protein. What is the difference in sequence between these multiple ThsB proteins? If you compare the sequences of the multiple genes (at the protein level) that are in the same region in a single strain against the structure, do you find interesting surface variability? What do TIR domains do in bacteria?

Bottom of page 5 – the specificity constant was determined and compared to other sirtuin proteins (Which ones? Are they bacterial? What are the functions of these other sirtuin containing proteins?) in the presence of acetyl peptide substrates, and is higher than deacetylated substrates. Does this tell us anything about the function of ThsA? Why is this point made? It should be made clear why this is important.

How were the oligos chosen for the EMSA assays? Was only one ssDNA and one dsDNA substrate tested? Were DprA and Smf shown to bind random DNA sequences? How was the length of the oligo chosen? As there is no understanding of how this system might interact with DNA, it's not clear to me that the lack of DNA binding shown in their experiment is meaningful.

What is the second Thoeris system that was characterized? Does the structure reveal anything interesting about this other system? How closely related are they?

How many copies of ThsB are found in *B. cereus*? Could other copies have the NAD⁺ cleavage activity?

The Rossmann-like fold of ThsB reveals structural similarity to a number of TIR domain proteins, but all listed are eukaryotic proteins. It would be more informative to address the microbial proteins that are most closely related.

Minor issues:

Fig. 1a – Why are errors bars shown only for the no protein experiment?

NAD cleavage in Fig 1 was assessed after 60 minutes. When the cleavage was assessed for the Asn112 and His152 mutants it was assessed at 30 minutes (Fig. 3d), and there is no WT control show. What is the NAD normalized to in this graph? Also, while the legend states 3 replicates \pm SEM, there are no error bars shown on the graph.

Fig. 3c, S7 – What do the boxes and coloured residues mean?

Reviewer #2 (Remarks to the Author):

Ka et al, mechanism of Thoeris defense system.

This work describes the structures of ThsA and ThsB, and proof that ThsA is an NAD⁺ cleaving enzyme. The structural and mutagenesis work confirms the previous suggestion that this protein cleaves NAD⁺ as part of a phage immune system. Here we have proof that the same mutation that was previously selected to inactivate phage immunity also abolished NAD⁺ cleavage by ThsA. The structural work is lovely and provides new structures for this system with thorough comparisons to similar folds and the appropriate analysis. I am left wondering, however, how this system actually works and I feel that paper doesn't quite answer that question.

The biochemical work is solid although seems to be missing pieces that the authors acknowledge they weren't able to re-capitulate in vitro. i.e. what does ThsB do? Does it interact with or modulate ThsA? Does it regulate activity in a different way? Additionally, it is unclear to me if the simple act of NAD⁺ cleavage is sufficient to induce abortive infection mechanisms. As the authors mentioned, expression of this protein in E.coli was non-toxic. I suggest that the authors consider trying to pulldown ThsA from E. coli perhaps during co-expression with ThsB. Does ThsA have other binding partners? Does co-expression make the situation toxic? Do NAD⁺ levels actually change in E. coli? Have the authors considered similar experiments but in the native host, Bacillus.

The key result here is the mutation that abolished immunity also abolished NAD⁺ cleavage, but this does not add too much to the overall story. I suppose it would be interesting if this weren't the case and the authors worked out another twist, but I worry that the addition beyond the structures does not give us a more clear picture of the mechanism of this system.

The suggestion by the authors that this may be an abortive system seems testable. This could be tested with phage infection and assessing cell survival.

Minor:

I would appreciate more background on what sirtuins are.

Reviewer #3 (Remarks to the Author):

Ka and colleagues present the first crystal structures of a novel bacterial immune system called Thoeris, consisting of the ThsA and ThsB proteins. They also show that ThsA is capable of cleaving NAD⁺, and that a mutation previously shown to disrupt Thoeris anti-phage activity resides within the ThsA active site for binding and cleaving NAD⁺. They propose that because this mutation in the active site blocks in vitro cleavage activity, that Thoeris relies on an NAD⁺ cleavage mechanism as part of its anti-phage activity. The structures of the SLOG domain of ThsA, and the ThsB proteins provide structural information to infer putative function, but activities for these features were not determined.

This is a well-written paper that is timely, as little is known regarding the mechanisms used by Thoeris. The structures and models are of high quality, and the biochemical assays appear to be performed well. The figures are excellent and the supplemental data improve the clarity of the paper. However, I feel the major conclusion of the paper, which is intriguing and could be correct, lacks the proper controls to be completely convincing. The conclusions of the authors should either be toned down, or more data should be included to support the claim that Thoeris relies on NAD⁺ cleavage to mediate anti-phage activity.

I have detailed one minor concern and one major concern below:

Minor:

On page 5 authors cite the *E. coli* intracellular NAD⁺ concentration to be within the same order of magnitude as the K_m value of ThsA, but it would be more appropriate to cite the concentration of NAD⁺ within *B. cereus*.

Major:

The authors present an interesting observation that the mutation N112A which previously blocked immune system function *in vivo*, also impairs NAD⁺ cleavage *in vitro*. The authors conclude that NAD⁺ cleavage is thus the anti-phage mechanism. I agree that the data suggest this is a possible mechanism. However, I do not believe enough controls have been made on this new system to confirm that these observations prove "that NAD⁺ degradation by ThsA is key to the antiphage function of Thoeris defense system". I suggest a few controls.

First, can the authors test to ensure that the mutant proteins maintain the same overall structure as wild type proteins? Circular dichroism of purified mutant proteins, or even crystal structures if available could confirm that mutation does not disrupt overall structure folding and activity.

Can the authors show that it is the degradation of NAD⁺ that is necessary for anti-phage activity, and not just binding? Do the point mutants bind NAD⁺ or is there a NAD⁺ binding defect? If no binding is observed, it cannot be ruled out that binding and not cleavage is what activates the immune system. A NAD⁺ binding assay would clarify how these point mutants are affecting activity. To support the statement that cleavage is key, I would like to see a mutant that disrupts cleavage but not binding, that when expressed *in vivo* disrupts the immune system.

We appreciate the constructive comments and suggestions from all three reviewers. Below is our point-by-point response to their concerns.

Reviewer #1:

(Comment 1-1) Thoeris system is found in >2000 microbial genomes –how diverse are the bacteria it's found in? Is it only strains of Bacillus, or is it widely distributed across different bacterial genus?

(Response) In the previous study by the Sorek group, the Thoeris system was found in more than 2000 sequenced microbial genomes with broad phylogenetic distribution. It was detected in nine different taxonomic phyla (Proteobacteria, Firmicutes, Bacteroidetes, Actinobacteria, Cyanobacteria, Euryarchaeota, Fusobacteria, Nitrospirae and Tenericutes). In the revised manuscript (Page 3, Lines 18 to 20), we have included the following sentences to read,

“.... with broad phylogenetic distribution. It was detected in nine different taxonomic phyla, including a wide variety of bacteria and archaea.”

We have also cited the previous study (ref. 14) by the Sorek group.

(Comment 1-2) In some cases ThsA has a transmembrane domain – how would this play into its activity? Where else are sirtuin (or Macro) domains found? In bacteria?

(Response) In some instances (~30%), ThsA has an N-terminal transmembrane domain, whereas the majority of ThsA homologues do not contain the N-terminal transmembrane domain but rather a C-terminal SLOG-like domain. Based on this and other findings, we carefully raised the possibility that the SLOG-like domain of ThsA substitutes the function of the transmembrane domain and is involved in the recognition and/or transmission of molecular signals. In Discussion (Page 14, Lines 5 to 11), we had included the following sentences to read,

“.... 30% of ThsA homologues do not contain the SLOG-like domain but rather an additional N-terminal multi-transmembrane domain. These findings suggest that the SLOG-like domain of ThsA is involved in the recognition and/or transmission of nucleotide-related or other molecular signals. Thus, the C-terminal domain of ThsA may play a role in the regulation or signaling associated with the NAD⁺ cleavage activity of the N-terminal domain. Nevertheless, the exact function of the SLOG-like domain in the Thoeris defense system awaits further investigation.”

We agree that more detailed explanations about the sirtuin-like and Macro domains and their bacterial members are needed in Introduction. In the revised manuscript (Page 3, Line 23 to Page 4, Line 9), we have included the following sentences to read, **“Sirtuins are a family of protein deacetylases whose activity is dependent on NAD hydrolysis. They are widely distributed from bacteria to higher eukaryotes. Its founding member, yeast Silent information regulator 2 (Sir2) is a histone deacetylase involved in a variety of cellular regulation. CobB is a well-characterized bacterial sirtuin that regulates the function of acyl-CoA synthetase by lysine deacetylation. The Macro domain is a widespread and conserved module of ~190 residues that can bind NAD metabolites or related molecules. It is named after the C-terminal domain of macroH2A, a variant of histone H2A containing an extensive C-terminal non-histone tail. In several pathogenic bacteria, the connections between Macro domains and pathogenesis have been suggested.”**

We have also cited appropriate references (refs. 16-21).

(Comment 1-3) The discussion around the ThsB protein should be fleshed out. In some cases, there are multiple ThsB proteins found around a single ThsA protein. What is the difference in sequence between these multiple ThsB proteins? If you compare the sequences of the multiple genes (at the protein level) that are in the same region in a single strain against the structure, do you find interesting surface variability? What do TIR domains do in bacteria?

(Response) In the *B. cereus* Thoeris system studied in our work, two *thsB* genes are found upstream and downstream of a single *thsA* gene. Their amino acid sequence identity is only 18%. Thus, it is difficult to recognize important surface variability. In the revised manuscript (Page 14, Line 22 to Page 15, Line 2), we have included the following sentences to read,

“In *B. cereus* MSX-D12, the second *thsB* gene is found upstream of the pair of *thsA* and *thsB* genes characterized in this study. The amino acid sequence identity is only 18% between the two ThsB homologues, indicating the diversity of multiple ThsB copies in Thoeris defense system.”

To provide additional information about bacterial TIR domains, we have included the following sentence in Introduction (Page 4, Lines 11 to 13) to read,

“Bacterial TIR-containing proteins have initially been hypothesized to function in pathogenesis but have also been implicated in other diverse mechanisms such as nucleic acid metabolism.”

(Comment 1-4) Bottom of page 5 – the specificity constant was determined and compared to other sirtuin proteins (Which ones? Are they bacterial? What are the functions of these other sirtuin containing proteins?) in the presence of acetyl peptide substrates, and is higher than deacetylated substrates. Does this tell us anything about the function of ThsA? Why is this point made? It should be made clear why this is important.

(Response) We apologize for the confusion. We wanted to compare the specificity constant of ThsA with those of other sirtuin family proteins to ensure the robustness of the enzymatic activity of ThsA. We thought this is important because ThsA lacks a structural element for zinc binding present in typical sirtuins, the disruption of which results in loss of the NAD⁺ hydrolysis function. The compared sirtuin proteins are all eukaryotic homologues including its founding member, yeast Sir2. We could not find references reporting the kinetic parameters for NAD⁺ hydrolysis by bacterial sirtuin homologues. We agree that the description of the deacetylation substrates is not essential but may rather cause unnecessary confusion. To make these points clear, we have modified the sentence (Page 6, Lines 19 to 20) to read,

“This falls within the range observed for NAD⁺ cleavage by eukaryotic sirtuin proteins including the yeast Sir2 (3.5 to 8205 M⁻¹·s⁻¹).”

We had also included the following sentence (Page 8, Lines 4 to 6) to read,

“ThsA also lacks the insertion element for zinc coordination present in sirtuins (Figs. 3a and Supplementary Fig. 4), the disruption of which results in loss of the NAD⁺ hydrolysis function of sirtuins.”

In Introduction of the revised manuscript (Page 3, Line 23 to Page 4, Line 9), we have also provided additional information about sirtuin proteins. Please see the second part of our response to the reviewers' comment 1-2.

(Comment 1-5) How were the oligos chosen for the EMSA assays? Was only one ssDNA and one dsDNA substrate tested? Were DprA and Smf shown to bind random DNA sequences? How was the length of the oligo chosen? As there is no understanding of how this system might interact with DNA, it's not clear to me that the lack of DNA binding shown in their experiment is meaningful.

(Response) In the previous study (Ref. 38), DprA and Smf interacted with DNAs of random sequence, indicating their affinity to non-specific DNAs. We agree that the lack of DNA binding by ThsA is not so meaningful, and having the EMSA analyses in the Results section may rather mislead readers. In the revised manuscript, we have relocated the sentences to Discussion (Page 13, Line 20 to 23) and have modified them to read,

“*Streptococcus pneumoniae* DprA and its *B. subtilis* ortholog (Smf) belong to the SLOG superfamily and have been previously shown to bind single-stranded DNA of random sequence. However, the DNA binding affinity was not observed for ThsA in our electrophoretic mobility shift assay (EMSA; Supplementary Fig. 14).”

(Comment 1-6) What is the second Thoeris system that was characterized? Does the structure reveal anything interesting about this other system? How closely related are they?

(Response) We appreciate the suggestion. In the revised manuscript, we have described the second characterized Thoeris system (*Bacillus amyloliquefaciens* Y2) with its difference from the *B. cereus* system studied in this paper. In Introduction (Page 4, Line 23 to Page 5, Line 3), we have included the following sentences to read, **“The other validated Thoeris system is from *Bacillus amyloliquefaciens* Y2. In this system, ThsA includes the Macro domain with the additional N-terminal transmembrane domain and a shorter C-terminal region compared to the *B. cereus* homologue.”**

For the discussion about the different ThsA domain structure between the two characterized Thoeris subtypes, please see the first part of our response to the reviewers' comment 1-2.

(Comment 1-7) How many copies of ThsB are found in *B. cereus*? Could other copies have the NAD⁺ cleavage activity?

(Response) In the revised manuscript (Page 14, Line 22 to Page 15, Line 2), we have indicated that two copies of ThsB are found in *B. cereus*. For more details, please see the first part of our response to the reviewers' comment 1-3.

As the reviewer pointed out, we cannot exclude the possibility that the uncharacterized second ThsB protein has NAD⁺ cleavage activity. However, we believe that this possibility does not affect the conclusion of this work since the second ThsB protein was dispensable for the antiphage function of the *B. cereus* Thoeris system. In the previous study by the Sorek group (Ref. 14), the single *thsA* gene and its downstream *thsB* gene encoding the first ThsB protein conferred resistance against phage infection when engineered into a model bacterium. The second *thsB* gene, found upstream of the *thsA* gene, was not required for the antiphage activity. To make this point clear, we have added the following sentence to the revised manuscript (Page 15, Lines 2 to 4) to read,

“Notably, the single *B. cereus thsA* gene and its downstream *thsB* gene were sufficient to confer antiphage resistance, suggesting the dispensability of the upstream *thsB* gene.”

(Comment 1-8) The Rossmann-like fold of ThsB reveals structural similarity to a number of TIR domain proteins, but all listed are eukaryotic proteins. It would be more informative to address the microbial proteins that are most closely related.

(Response) As the reviewer indicated, we listed the top ten structural neighbors of the Rossmann-like fold of ThsB in Supplementary Table 3. However, they are NOT all eukaryotic proteins. Two of them are bacterial proteins. They are a nucleoside 2'-deoxyribosyltransferase from *Bacillus psychrosaccharolyticus* and a small GTP-binding protein from *Thermosiphon melanesiensis*. In the revised manuscript, we have marked the bacterial entries with asterisks in Supplementary Table 3.

In Supplementary Fig. 13, we also provided the structural comparison of ThsB with two bacterial enzymes, a CMP hydrolase from *Streptomyces rimofaciens* and a nucleoside 2-deoxyribosyltransferase from *Lactobacillus leichmannii*. To make this point clear, we have modified the legend of Supplementary Fig. 13 to read, **“... b, Structural comparison of ThsB and bacterial nucleotide hydrolases. ThsB (red) is structurally aligned with the CMP-bound MilB (PDB ID: 4JEM; green), a CMP hydrolase from *Streptomyces rimofaciens*, and 5-methyl-2'-deoxypseudouridine (5MD)-bound nucleoside 2-deoxyribosyltransferase (PDB ID: 1F8Y; yellow) from *Lactobacillus leichmannii*.”**

Minor issues:

(Comment 1-9) Fig. 1a – Why are error bars shown only for the no protein experiment?

(Response) We did show error bars for all proteins, but some of them were too small to recognize. We apologize for this. In the revised manuscript, we have modified Fig. 1a to show the error bars more clearly. With the revised manuscript, we have also provided the source data file that contains the raw data underlying all reported averages in the graphs.

(Comment 1-10) NAD cleavage in Fig 1 was assessed after 60 minutes. When the cleavage was assessed for the Asn112 and His152 mutants it was assessed at 30 minutes (Fig. 3d), and there is no WT control shown. What is the NAD normalized to in this graph? Also, while the legend states 3 replicates \pm SEM, there are no error bars shown on the graph.

(Response) We measured NAD⁺ cleavage for the mutant proteins until 30 min because the wild type ThsA depleted almost all of the NAD⁺ substrates within 10 min (Fig. 1a). Thus, we believe that the 30 min data are sufficient to contrast the loss-of-function mutants with the wild type protein. To make this point clear, we have shown again the wild type data (Fig. 1a) as a control for comparison in Fig. 3d of the revised manuscript. Accordingly, we have changed the legend of Fig. 3d to read, **“... The graph for the wild type ThsA (Fig. 1a) are shown again as a control for comparison.”**

The amounts of remaining NAD⁺ were normalized to the 0 min data for each protein. We have included the following sentence in the legend of Fig. 3d to read, **“Data are normalized to the values measured at 0 min for each protein”**

In the revised manuscript, we have modified Fig. 3d to display the error bars more clearly. With the revised manuscript, we have also provided the source data file that contains the raw data underlying all reported averages in the graphs.

(Comment 1-11) Fig. 3c, S7 – What do the boxes and coloured residues mean?

(Response) We appreciate the suggestion. In the revised manuscript, we have included the following sentences in the legends of Fig. 3c and Supplementary Fig. 6 to read,

“White character in red box indicates strict identity. Red character and blue frame represent similarity in a group and across groups, respectively.”

Reviewer #2:

(Comment 2-1) The biochemical work is solid although seems to be missing pieces that the authors acknowledge they weren't able to re-capitulate in vitro. i.e. what does ThsB do? Does it interact with or modulate ThsA? Does it regulate activity in a different way?

(Response) We tested the interaction between ThsA and ThsB, but found that ThsB did not interact with ThsA in the analytical SEC experiment (Supplementary Fig. 1b, c). We had included the following sentences (Page 10, Line 19 to Page 11, Line 2) to read,

“We also suspected the possibility that ThsB may function as a component of a larger multi-protein complex formed with ThsA. However, in our analytical SEC experiment, ThsB was eluted separately from ThsA with and without NAD⁺ (Supplementary Fig. 1b, c). These observations suggest that ThsB does not form a stable complex with ThsA.”

We also explored the possibility that ThsB regulates or inhibits the NAD⁺ cleavage activity of ThsA. In our experiment, ThsA was still active in the presence of ThsB, and the cleavage products (Nam and ADPR) were identical with and without ThsB (Supplementary Fig. 11). This suggests that ThsB does not alter the function of ThsA. To make this point clear in the revised manuscript (Page 10, Lines 16 to 19), we have modified the following sentence to read,

“It is also unlikely that ThsB alters the function of ThsA or catalyzes the potential pathway(s) downstream of NAD⁺ cleavage because ThsA was still active in the presence of ThsB, which did not further process the cleavage products Nam and ADPR (Supplementary Fig. 11).”

We have also revised the legend of Supplementary Fig. 11 to read,

“Supplementary Fig. 11. ThsB does not alter NAD⁺ cleavage function of ThsA or further degrade NAD⁺ cleavage products. in the presence of ThsB, ThsA is still functional, and the cleavage products (Nam and ADPR) remain intact.”

(Comment 2-2) Additionally, it is unclear to me if the simple act of NAD⁺ cleavage is sufficient to induce abortive infection mechanisms. As the authors mentioned, expression of this protein in E.coli was non-toxic. I suggest that the authors consider trying to pulldown ThsA from E. coli perhaps during co-expression with ThsB. Does ThsA have other binding partners? Does co-expression make the situation toxic? Do

NAD⁺ levels actually change in *E. coli*? Have the authors considered similar experiments but in the native host, *Bacillus*.

(Response) As the reviewer suggested, we have tested the co-expression of ThsA and ThsB. In the experiment, (His)₆-MBP-tagged ThsA was expressed together with untagged ThsB in *E. coli* cells, but the untagged ThsB did not co-purify with the histidine-tagged ThsA in nickel affinity chromatography (Supplementary Fig. 12), suggesting that they do not form a stable complex. In the revised manuscript (Page 10, Line 22 to Page 11, Line 2), we have included the following sentences to read,

“When expressing (His)₆-maltose binding protein (MBP)-tagged ThsA together with untagged ThsB in *E. coli* cells, ThsB did not co-purify with ThsA in nickel-affinity chromatography (Supplementary Fig. 12). These observations suggest that ThsB does not form a stable complex with ThsA.”

The co-expression of ThsA and ThsB did not result in toxicity in *E. coli* cells, either. In the revised manuscript (Page 15, Lines 17 to 18), we have modified the following sentence to read,

“During the expression of recombinant ThsA and/or ThsB in *E. coli*, we did not observe growth arrest or toxicity.”

It is also unclear to us whether the NAD⁺ cleavage by the Thoeris system is sufficient to cause cell death. Thus, in Discussion (Page 15, Lines 15 to 20), we had included the following sentences to read,

“It is unclear whether the NAD⁺ cleavage by the Thoeris system is sufficient to cause cell death by NAD⁺ depletion. This suggests that the Thoeris system does not deplete the intracellular NAD⁺ to drive direct cell death and is a part of a more elaborate defense mechanism requiring downstream signaling after NAD⁺ cleavage as in the case of the plant TIR NAD⁺ hydrolases.”

We appreciate the suggestion about the additional experiments using *E. coli* or *B. cereus*. Although we are seriously considering doing such experiments, we also feel that they merit another publication, and our current data are sufficient to draw meaningful conclusions.

(Comment 2-3) The key result here is the mutation that abolished immunity also abolished NAD⁺ cleavage, but this does not add too much to the overall story. I suppose it would be interesting if this weren't the case and the authors worked out another twist, but I worry that the addition beyond the structures does not give us a more clear picture of the mechanism of this system.

(Response) We agree that the two novel crystal structures are the highlights of this paper, but respectfully disagree with the reviewer on the value of our additional functional data. We believe that the results from our functional analyses are crucial to significantly advance the understanding of the Thoeris defense mechanism. For example, prior to this work, it has not been experimentally proven whether the Thoeris system actually functions with NAD⁺. It was also not clear which one of the two Thoeris proteins, ThsA or ThsB, is responsible for NAD⁺ cleavage. To make this point clear, in the last summarizing paragraph of Introduction (Page 5, Lines 7 to 9), we have included the following sentence to read,

“We found that ThsA exhibited robust NAD⁺ cleavage activity. Combined with previous findings, these results advance our understanding of the molecular mechanism underlying the Thoeris defense system....”

(Comment 2-4) The suggestion by the authors that this may be an abortive system seems testable. This could be tested with phage infection and assessing cell survival. (Response) We agree that the suggestion is testable. Without the test result, we think that we can still raise the possibility in the Discussion section, but, at the same time, the suggestion need to be toned down. In Discussion of the revised manuscript (Page 15, Lines 13 to 14), we have modified the following sentence to read, **“Based on these observations, we carefully raise the possibility that the Thoeris system is an “altruistic suicide” defense system”**

Minor:

(Comment 2-5) I would appreciate more background on what sirtuins are. (Response) We appreciate the suggestion. In Introduction of the revised manuscript (Page 3, Line 23 to Page 4, Line 5), we have included the following sentences to read, **“Sirtuins are a family of protein deacetylases whose activity is dependent on NAD hydrolysis. They are widely distributed from bacteria to higher eukaryotes. Its founding member, yeast Silent information regulator 2 (Sir2) is a histone deacetylase involved in a variety of cellular regulation. CobB is a well-characterized bacterial sirtuin that regulates the function of acyl-CoA synthetase by lysine deacetylation.”**

We have also cited appropriate references (refs. 16, 18, 19).

Reviewer #3:

I have detailed one minor concern and one major concern below:

Minor:

(Comment 3-1) On page 5 authors cite the *E. coli* intracellular NAD⁺ concentration to be within the same order of magnitude as the Km value of ThsA, but it would be more appropriate to cite the concentration of NAD⁺ within *B. cereus*.

(Response) We agree with the reviewer. Indeed, we had also tried to cite a reference reporting the intracellular NAD⁺ level of *B. cereus*, but could not find one. Instead, we were able to find the value for another bacterium (*E. coli*). To make this point clear, we have modified the following sentence (Page 6, Line 21) to read, **“.... the intracellular NAD⁺ level of a bacterium *Escherichia coli* was previously estimated”**

Major:

(Comment 3-2) The authors present an interesting observation that the mutation N112A which previously blocked immune system function in vivo, also impairs NAD⁺ cleavage in vitro. The authors conclude that NAD⁺ cleavage is thus the anti-phage mechanism. I agree that the data suggest this is a possible mechanism. However, I do not believe enough controls have been made on this new system to confirm that these observations prove “that NAD⁺ degradation by ThsA is key to the antiphage function of Thoeris defense system”. I suggest a few controls.

(Response) We agree that our conclusion is a bit strong and needs to be toned down although consistent with our data. In the revised manuscript, we have toned down the claim that NAD⁺ degradation by ThsA is key to the antiphage function of Thoeris defense system. In Abstract (Page 2, Lines 7 to 8) and Results (Page 10, Lines 2 to 4), we have modified the following sentences to read,

“Mutation analysis suggests that NAD⁺ cleavage is linked to the antiphage function of Thoeris.” and **“our mutation analysis suggests that NAD⁺ degradation by ThsA is linked to the antiphage function of the Thoeris defense system.”**, respectively.

We have also modified a subheading in Results (Page 8, Line 21) to read, **“NAD⁺ cleavage by ThsA is linked to antiphage activity”**

(Comment 3-3) First, can the authors test to ensure that the mutant proteins maintain the same overall structure as wild type proteins? Circular dichroism of purified mutant proteins, or even crystal structures if available could confirm that mutation does not disrupt overall structure folding and activity.

(Response) We appreciate the suggestion. As suggested, we performed circular dichroism (CD) spectroscopy for the wild type and mutant ThsA proteins (Supplementary Fig. 7). All of the wild type and mutant proteins exhibited CD spectra typical of folded proteins (Supplementary Fig. 7). However, our analytical SEC analyses (Supplementary Fig. 8a) indicated that the oligomeric state of ThsA is disrupted by the H152A mutation, but not by the N112A mutation. Thus, in the revised manuscript, we have drawn a conclusion only for the N112A mutant that retains the overall folding of the wild type as well as its oligomeric state. In Results (Page 9, Lines 12 to 23), we have included the following sentences to read, **“Both of the purified ThsA mutants, N112A and H152A, exhibited circular dichroism (CD) spectra similar to that of the wild type (Supplementary Fig. 7), which is characteristic of folded proteins. In analytical SEC, the chromatogram of the N112A mutant was essentially identical to that of the wild type control, but the H152A mutant was eluted significantly later (Supplementary Fig. 8a). These results suggest that the N112A mutant retains the overall folding of the wild type as well as its oligomeric state, but the H152A mutation disrupts the proper oligomerization of ThsA. Based on these observations, we concluded that the Asn112 residue is critical for NAD⁺ hydrolysis of ThsA. For the H152A mutant, it cannot be ruled out that the inactivation is caused by the disruption of the functionally relevant oligomeric state.”**

(Comment 3-4) Can the authors show that it is the degradation of NAD⁺ that is necessary for anti-phage activity, and not just binding? Do the point mutants bind NAD⁺ or is there a NAD⁺ binding defect? If no binding is observed, it cannot be ruled out that binding and not cleavage is what activates the immune system. A NAD⁺ binding assay would clarify how these point mutants are affecting activity. To support the statement that cleavage is key, I would like to see a mutant that disrupts cleavage but not binding, that when expressed in vivo disrupts the immune system.

(Response) We appreciate this critical comment. We carried out the suggested NAD⁺ binding assay using analytical SEC (Supplementary Fig. 8b), in which the ThsA mutants did not bind NAD⁺. In the revised manuscript (Page 9, Lines 18 to 19), we have included the following sentence to read,

“In subsequent binding assays using analytical SEC, neither of the mutants bound NAD⁺ (Supplementary Fig. 8b).”

Thus, as the reviewer pointed out, it is possible that the binding to NAD⁺, not its cleavage, is important for the antiphage function, and we have discussed this possibility in the revised manuscript. In Discussion (Page 13, Lines 10 to 12), we have included the following sentence to read,

“It cannot be completely ruled out that the binding to NAD⁺, not its cleavage, is important for the antiphage function since the N112A mutant did not bind NAD⁺.”

We have also toned down our statements. Please see our response to the reviewers' comment 3-2.

REVIEWERS' COMMENTS:

Reviewer #3 (Remarks to the Author):

The authors have addressed my previous concerns. I believe this manuscript will be of interest to the readers of Nature Communications.